

# Effects of sample size on estimation of rainfall extremes at high temperatures

Berry Boessenkool[1], Gerd Bürger[1], and Maik Heistermann[1]

[1]Potsdam University, Institute for Earth and Environmental Sciences, Karl-Liebknecht-str. 24, Haus 1, 14476 Potsdam Golm, Germany

*Correspondence to:* Berry Boessenkool (boessenk@uni-potsdam.de)

**Abstract.** High precipitation quantiles tend to rise with air temperature, following the so-called Clausius-Clapeyron scaling. This CC-scaling relation breaks down, or even reverts, for very high temperatures. In our study, we verify this reversal using a 60-year period of summer data in Germany. One of the suggested meteorological explanations is limited moisture supply, but our findings indicate that this behavior could also originate from simple undersampling. The number of observations in high temperature ranges is small, so extreme rainfall intensities following CC-scaling may not yet be recorded but logically possible. Because empirical quantile estimators using plotting positions drop with decreasing sample size, they cannot correct for this effect.

By fitting distributions to the precipitation records and using their parametric quantile, we obtain estimates of rainfall intensities that continue to rise with temperature. This procedure requires far fewer values (ca 50 for the 99.9 % quantile) to converge than classical order based empirical quantiles (ca 700). From the evaluation of several distribution functions, the Wakeby distribution appears to capture the precipitation behavior better than the General Pareto Distribution (GPD). Despite being parametric, GPD estimators still show some underestimation in small samples.

*Keywords: extreme precipitation intensity, Clausius-Clapeyron scaling, parametric quantile estimators*

## 1 Introduction

The atmospheric water holding capacity and thus potential precipitation intensity and (flash) flood risk depends exponentially on temperature according to the Clausius-Clapeyron relationship (CC).

As empirically documented by several studies, high precipitation quantiles (Pq) rise with temperature (T), increasingly so with shorter time spans, such as hourly or smaller. This CC-scaling thus describes a log-linear dependence of Pq on T that roughly follows the CC-rate of 7 %/K for water vapor; T is some measure of temperature at the time of the event, mostly daily. Similarly well documented is a breakdown or even reversal of that relation for temperatures beyond some threshold, usually somewhere near 15 to 20 °C, as indicated in Fig. 1. This drop is also observed by Brandsma and Buishand (1997), Klein Tank and Koennen (1993), Panthou et al. (2014) and Westra et al. (2014).

Several explanations for this phenomenon have been proposed, such as an increase in the proportion of rainfall stemming from convective events as opposed to large scale stratiform precipitation (Haerter and Berg, 2009). Other explanations include





a slower increase of moisture availability than in moisture storage capacity according to the CC-relationship (Berg et al., 2009) or fully-saturated conditions lasting shorter than event duration (Hardwick Jones et al., 2010). There may be several different mechanisms in process at different time scales and locations (Utsumi et al., 2011).

The decrease in precipitation intensity at high temperatures coincides with a decrease in the number of observations. The aim of this study is to examine whether this drop could (partly) be a statistical sample size artefact. We obtained time series of precipitation and temperature from 14 stations across Germany from the German Weather Service (DWD). Each dataset contains data from May to September, 1951–2010. The hourly rainfall depth is recorded by Hellmann gauges with a resolution of 1/10 mm. The daily temperature average is computed from hourly measurements recorded at a height of 2 m above ground level. The plots to illustrate our analysis refer to Potsdam in the northeast of Germany (52°23'N, 13°04'E, 81 m above sea level). Judged on the basis of histograms of precipitation records, Potsdam is representative of the 14 stations.

## 2 Empirical quantile estimation

With a set of simulation experiments, we investigate our hypothesis that precipitation quantiles drop at high temperatures due to a smaller sample size. Random samples of different sizes are drawn from a) the full dataset and b) a synthetic Pq-T-relationship continuously rising with T. With these samples, we compare direct empirical quantiles with parametric quantiles from fitted distributions. We also quantify the effect of the fitting procedure on the shapes and positions of the distributions. The empirical quantiles are computed as the average of the results returned by nine different methods to obtain consecutive order statistics (as implemented in the software package R).

Following the analysis method of Lenderink and Meijgaard (2008) and Berg and Haerter (2013), we partition the hourly precipitation depths according to the daily mean air temperature. We use temperature bins with a fixed width of two degrees Celsius. Bin midpoints increase in 0.1 degree steps. The empirical precipitation quantiles per bin are presented in Fig. 2. The form of the Pq-T relationships is consistent with the behavior mentioned in the introduction section and similar in all 14 stations.

The Clausius-Clapeyron governed saturation vapor pressure is calculated with the August-Roche-Magnus approximation as specified by Hardwick Jones et al. (2010):

$$VPsat = e_s = 6.1094 \cdot exp\left(\frac{17.625 \cdot temp}{temp + 243.04}\right) \tag{1}$$

The derivation of Eq. (1) yields a CC-rate for rainfall intensity change of 7 % per degree at 0 °C and 6 %/K at 20 °C.

$$\frac{d\,e_s}{d\,T} = \frac{26170.1}{(T + 243.04)^2} \cdot exp\left(\frac{17.625 \cdot T}{T + 243.04}\right) \tag{2}$$

Along the CC-lines in Fig. 2, it can be estimated that the rate of precipitation increase follows CC-scaling for low temperatures and shows super-CC-scaling between 15 and 20 °C.





## 3 Distribution fitting

We fit 17 distribution functions to the datasets to examine which type fits best, using code developed in the R package extreme-Stat (Boessenkool, 2016). The parameters are estimated via linear moments. These are analogous to the conventional statistical moments (mean, variance, skewness and kurtosis), but "robust [and] suitable for analysis of rare events of non-Normal data.
[...] L-moments are especially useful in the context of quantile functions" (Asquith, 2016, 2011; Hosking, 1990).

Instead of applying the block maxima approach, we want to use all hourly records. Measurements of low rainfall intensities have a higher relative uncertainty and fitting to all the non-zero values misses tail properties completely, thus values below 0.5 mm/h are omitted. The goodness of fit is judged by the root mean square error (RMSE) of the points in the (empirical) cumulated density functions of the sample and the distribution functions.

If the lower values are truncated first (Peak Over Threshold approach), the resulting censored quantiles are more robust. With increasing threshold, the different distribution functions are located closer to each other. In this setting, probabilities must be adjusted. For example, if the censored Q0.99 is to be computed from the top 20 % of the data, Q0.95 of the truncated sample must be used.

Berg et al. (2009) used the top 20 % of precipitation intensities to fit a General Pareto Distribution (GPD or gpa). Lenderink
and Meijgaard (2008) used the top 5 %. (See also Table 1). As opposed to Haerter et al. (2010), we find that GPD (and Kappa) parametric quantiles rise with truncation percentage, see Fig. 3. Weibull (and Pearson III) quantiles slightly decrease, while Wakeby quantiles are stable. GPD-estimates are stable in the R packages or methods using maximum likelihood estimation (MLE) instead of probability weighted moments (PWM) or linear moments (LM), see Fig. 4.

We therefore use a weighted average of the 17 distribution function quantile estimates. Weights are based on the goodness
of fit of the distribution functions to the empirical data across each temperature bin at each of the 14 stations. For the following simulations we use a truncation proportion of 80 %. Because at least 5 values are needed to estimate the distribution parameters, the minimum sample size for this study is 25. To quantify the behavior of quantile estimators in (very) small samples, we run 2000 simulations described in the next section.

## 4 Sample size dependency simulations

For each sample size (n) between 25 and 800, as well as 1k and 2k, 2000 samples are taken from the 7667 precipitation values. From these samples, empirical and parametric quantiles are estimated. The median results are shown in Fig. 5. Due to the inherent structural differences between distribution functions, parametric quantile estimates range from 20 to 40 mm/h, thus the weighted average is slightly higher than the empirical value of the full dataset.

Empirical quantiles at small sample sizes reach the actual value of the full dataset in only a few simulations and converge
to this value asymptotically only at sample sizes larger than 700. This plot indicates that small sample sizes could indeed be a reason for low empirical quantiles at high temperature bins, as those usually contain few values (see the sample sizes in Fig. 2).





The weighted average of parametric quantiles does not systematically depend on sample size. The random error is larger than that of the empirical quantiles (the uncertainty range is spread more widely). They can sometimes still underestimate the actual quantile, but the systematic bias is eliminated.

The choice of distribution may have a large effect, especially at high quantiles. The 99.99 % quantiles for the Wakeby, Kappa,
Weibull and General Pareto distributions are 42, 42, 55, and 32 mm, respectively (37, 23, 77, and 14 if not truncated before fitting). Estimating this with empirical quantiles would require a sample size of at least 10 000; the analysis included 7667 values. Compared to the observed maximum of 39 mm in the last 60 years, the Weibull distribution appears to overestimate very high precipitation in this particular dataset, whereas the General Pareto distribution underestimates it. This may be dependent on the particular GPD implementation used, as seen for the 99.9 % quantile in Fig. 6.

In order to demonstrate that small sample sizes can actually reduce empirical precipitation quantile estimates at high temperatures, we define a synthetic temperature dependent Wakeby distribution following CC-scaling. Its five parameters (see left panels in Fig. 7) are derived from linear regression of the parameters estimated per temperature bin.

For each bin, a random sample with the size of the original sample in this bin is generated and the empirical and parametric 99.9 % quantiles are calculated. The resulting quantile estimations from 2000 simulations are aggregated in the right panel of
Fig. 7.

Even though the distribution continues to increase with temperature, empirical quantiles from random samples stagnate or drop around 20 °C where sample size decreases quickly. Parametric quantiles obtained by distribution fitting do not drop much. (The slight decrease stems from the highly weighted Kappa distribution, others show almost no drop.) Due to the greater uncertainty of parametric quantile estimates, a single random observation may also sometimes show a decrease at high
temperatures.

## 5   Discussion

The procedure of obtaining parametric and empirical quantiles was applied per temperature bin to all 14 stations (Fig. 8). Between 20 and 26 °C, where the empirical values decrease, the parametric quantiles keep increasing. This difference is less pronounced for quantiles below 99 %.

The observed precipitation quantile drop at high temperatures can be due to small sample sizes at high temperatures. Alternative explanations considering meteorological processes should not lightly be discarded however. Some were summarized briefly in Sect. 1. It might also, for example, be hypothesized that near-surface temperature is not an adequate proxy for air temperature at the height where precipitation forming patterns unfold on very warm days.

The distribution fitting procedure allows and mandates several choices.

1) It must be defined what minimum measurement value constitutes precipitation, below which the values are cut off. After testing several cutoff points, we choose 0.5 mm/h as a good compromise between data originality and quantity on one hand, and fit quality on the other. With a larger cutoff, a larger proportion of the data is discarded, and the empirical 99 % quantile increases from 9 to 18 mm (at 0.1 and 2 mm cutoff) and distribution fit rankings change.





2) To select distribution functions, their goodness of fit must be determined. RMSE is a good measure for the closeness of distribution functions to the empirical distribution of the actual values, but other measures are available as well.

3) In a Peak Over Threshold (POT) approach, the distributions are fit closely to high values. This is useful, as the observed extremes are especially important for high quantile estimation. The choice of threshold value or truncation percentage affects distribution ranking and quantile estimate. With higher thresholds, the different estimates appear to converge, but necessary sample size increases.

4) Threshold excesses for large samples and thresholds should converge to the GPD. However, for small samples other distribution functions may fit better. Fitting the GPD with MLE is stable with regard to truncation percentage, but more dependent on sample size than methods using PWM or linear moments. Different software implementations also yield different quantile estimates.

The parametric method requires significantly fewer data points in a sample than empirical quantiles need to converge to the actual (unknown) value. In the combination of small sample sizes and very high quantiles, it is recommendable to use parametric quantiles.

## 6   Conclusions

The increase in rainfall intensity with temperature is relevant for local flood-risk computation, for example in urban drainage system development. Precipitation quantile estimates rise with temperature until a turning point beyond which they decrease. Besides possible meteorological limitations, simulations indicate that this can be due to sample size dependency in case empirical quantiles are used. The types of simulations described here could be a useful method to determine the necessary sample size per bin.

Parametric quantiles from fitted distributions provide a means to retrieve less systematically biased estimates of extreme quantiles. The random error is larger, but this effect can be quantified by confidence intervals obtained by bootstrapping. The quantile estimates are sensitive to the distribution fitting (parameter estimation) procedure. Using "out-of-the-box" GPD quantile estimates may not suffice to reduce the effect of sample size.

## 7   Data availability

The contract with DWD regarding the exchange of data does not allow us to publish raw data. However, shorter time series (up to 20 years) of hourly rainfall observations can be obtained from ftp://ftp-cdc.dwd.de/pub/CDC/observations_germany/ climate/hourly/precipitation/historical

*Author contributions.*  B. Boessenkool: analysis and manuscript. G. Bürger & M. Heistermann: original idea, guidance, review

*Acknowledgements.*  We wish to thank DWD for preparing and providing the datasets, as well as William Asquith for reviewing our manuscript before submission.



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





**Table 1.** PT-analysis methods used in the literature. (p): personal communication per email.

| Article | bin width | min $n_{bin}$ | quantile estimation method |
|---|---|---|---|
| Klein Tank and Koennen (1993) | 2 °C | unknown | mean amount |
| Brandsma and Buishand (1997) | 2 °C | unknown | mean amount |
| Lenderink and Meijgaard (2008) | 2 °C | unknown | emp. + GPD top 5 and 10 % |
| Berg et al. (2009) | 2 °C | 300 | GPD top 20 %; mm/day (for each month) |
| Hardwick Jones et al. (2010) | variable | median 233 | empirical order stats (p) |
| Lenderink et al. (2011) | 2 °C (overlap: 1 °C steps) | 200 (p) | empirical + GPD top 4 %; as in climexp.knmi.nl (p) |
| Utsumi et al. (2011) | var., avg. 2 °C | 150 | unknown, presumably empirical |
| Berg and Haerter (2013) | 5 °C (overlap: 3 °C steps) | 300 | unknown, presumably empirical |
| Berg et al. (2013) | 1 °C | unknown | unknown, presumably empirical |
| Panthou et al. (2014) | 2 °C (overlap: 1 °C steps) | 100 | empirical; Cunnane unbiased estimator (p) |
| Westra et al. (2014) | - | - | as in Lenderink et al. (2011) |

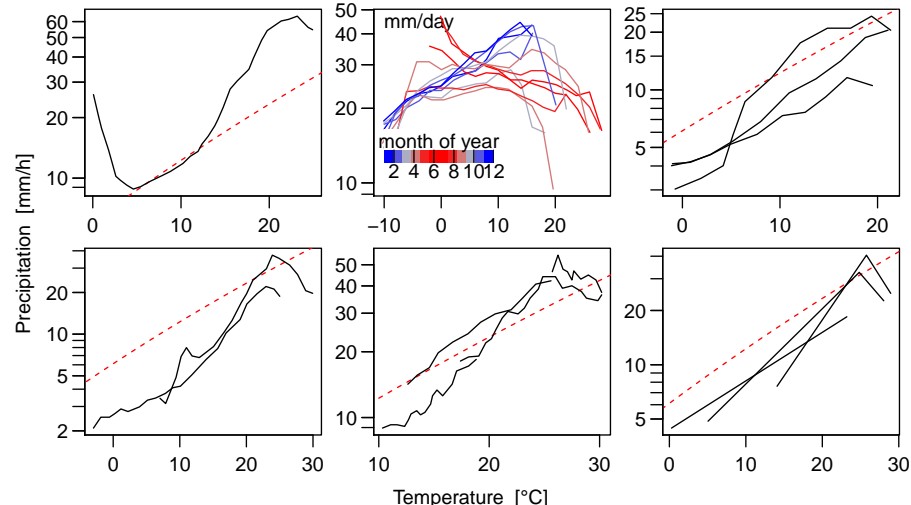

**Figure 1.** Pq-T-Relationships (99 % quantile, hourly intensities) digitized from several figures in the literature on a logarithmic scale. Red dashed lines indicate CC-scaling (see Sect. 2). Across regions and studies, Pq rises with T but then decreases. *Top row*: Berg et al. (2013), Berg et al. (2009) (mm/day), Berg and Haerter (2013). *Bottom row*: Lenderink et al. (2011), Hardwick Jones et al. (2010), Utsumi et al. (2011) (converted from mm/day). The last two articles use temperature bins of varying width with semi-constant number of observations per bin.




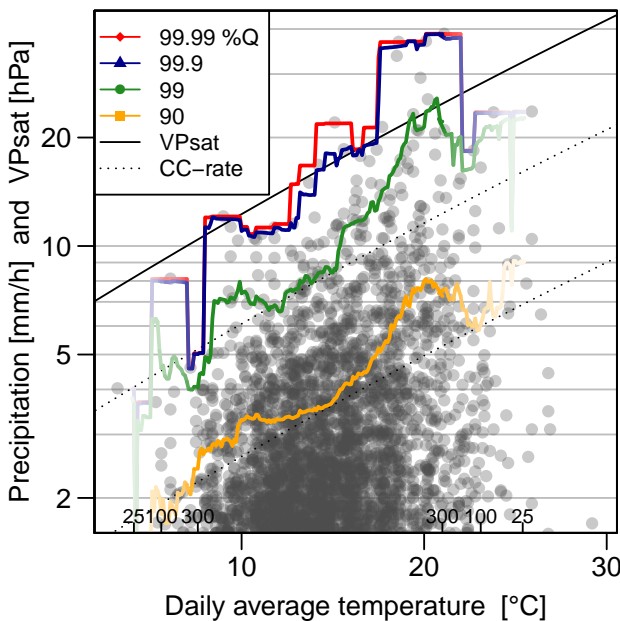

**Figure 2.** Empirical quantiles of hourly precipitation intensities per temperature bin in Potsdam on a logarithmic scale. $e_s$ and CC-scaling are drawn as reference lines. The numbers at the bottom indicate the temperature where the number of data points per bin exceed 25, 100 and 300.

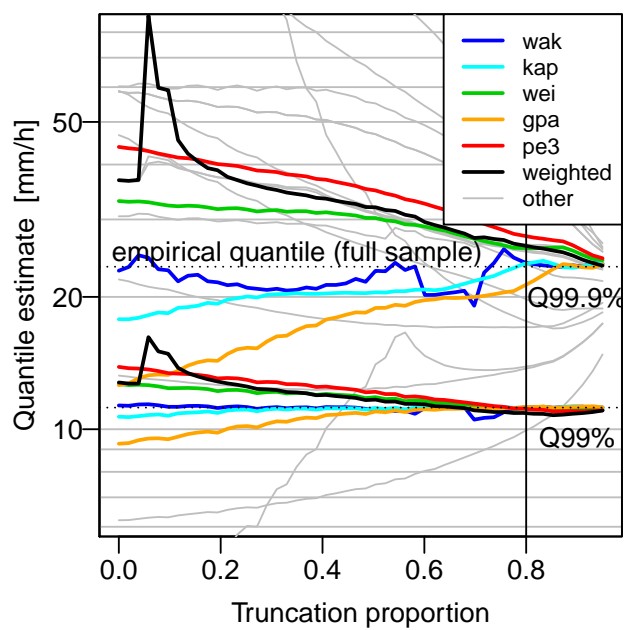

**Figure 3.** Dependency of the parametric 99.9 % quantile estimate on the truncated proportion (for the complete dataset with 8k precipitation records). For selected distributions, the 99 % quantiles are also shown.





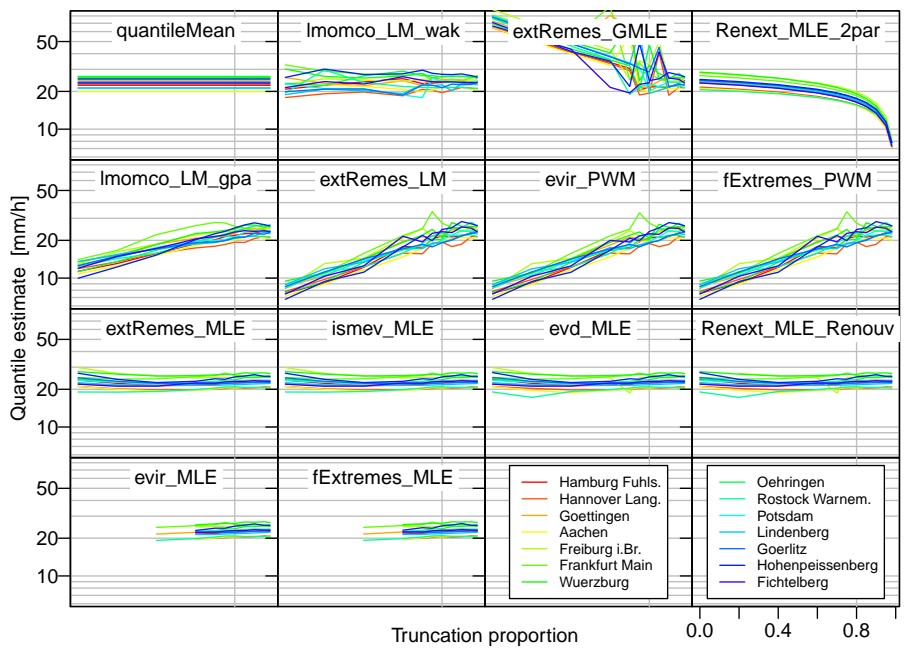

**Figure 4.** 99.9 % precipitation intensity quantile of the complete dataset of each station, depending on the proportion of the data truncated, for different R packages and methods. Most of the estimates are obtained with the function q_gpd in the R package extremeStat. MLE: Maximum Likelihood Estimation, PWM: Probability Weighted Moments, LM: Linear Moments

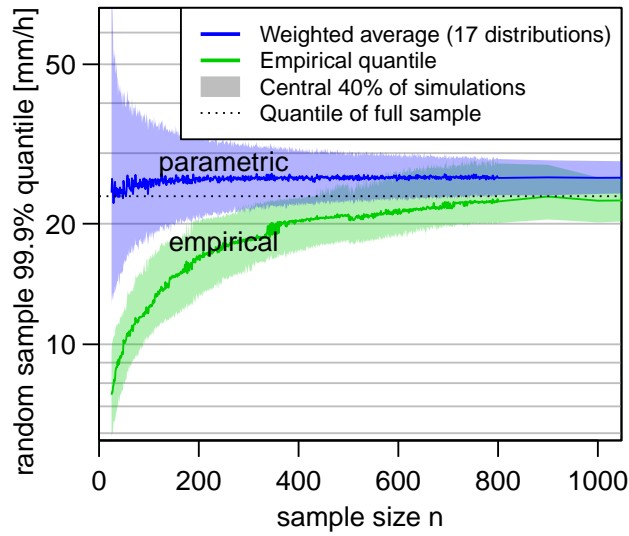

**Figure 5.** Dependency of empirical and parametric 99.9 % quantile on the size of samples drawn from all the 8k precipitation intensity values observed in Potsdam. The horizontal dashed line marks the empirical quantile of the complete dataset.




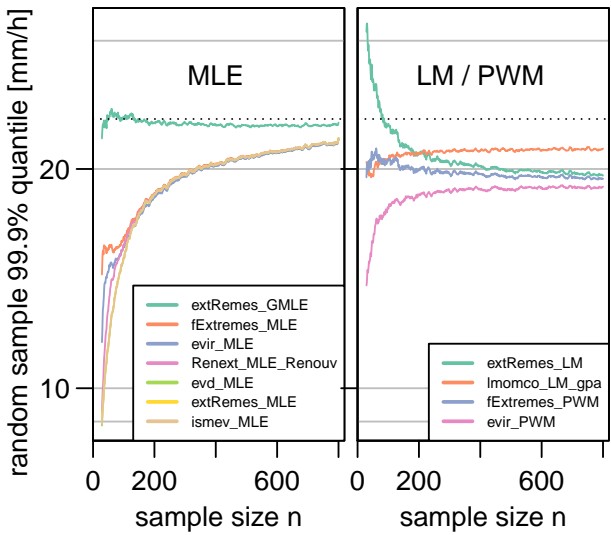

**Figure 6.** Dependency of GPD 99.9 % quantile estimates on sample size. Computed with 7 different R packages using MLE, PWM or LM. By fundamental theory, fitting via PWM and LM yields the same distribution parametrizations.

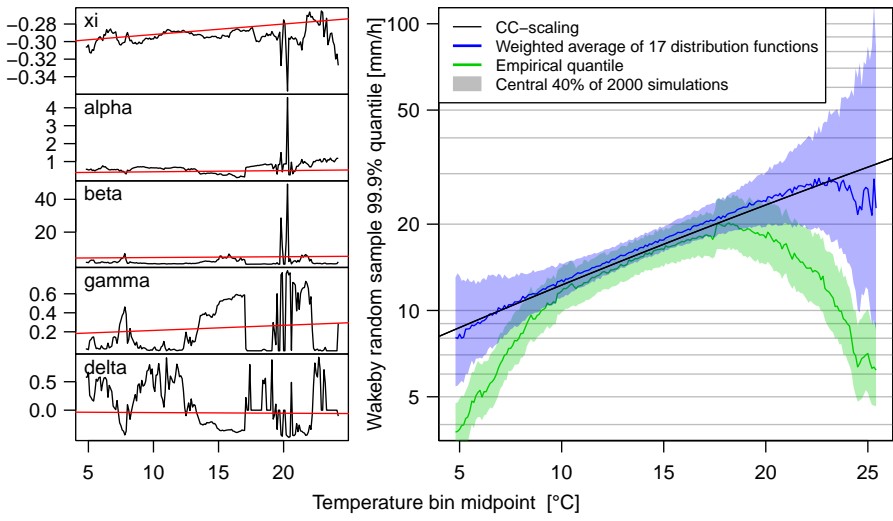

**Figure 7.** *Left panels*: Parameters of a temperature-dependent Wakeby distribution. *Right panel*: Corresponding 99.9 % distribution quantile (black line at CC-scaling) and quantiles generated from random samples in 2000 simulations. The blue line shows the median of empirical, the green line of parametric quantiles.





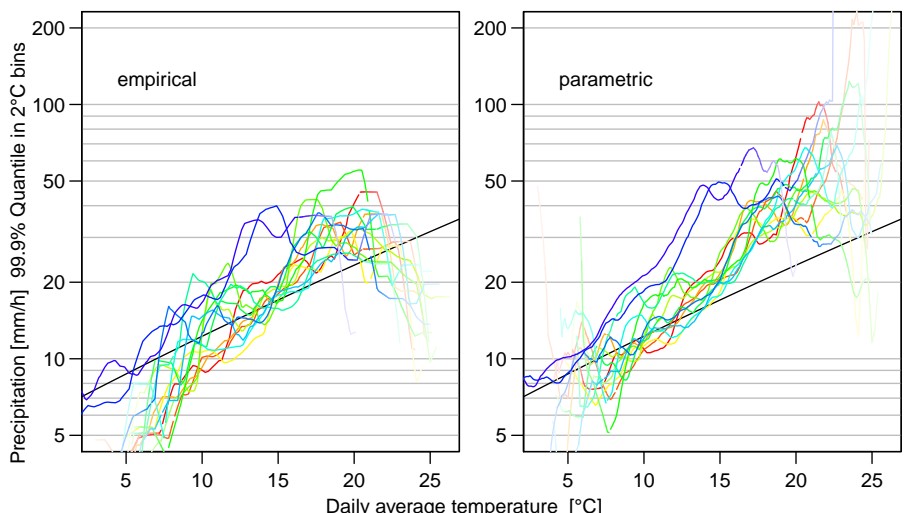

**Figure 8.** 99.9 % precipitation intensity quantile per temperature bin. Colors denote the 14 stations, Potsdam is light green. The lines are smoothed with a 9-point (=1 °C) moving average to show more signal instead of noise. Colors fade away as the number of observations per bin decreases. The two blue stations are located on mountains.