# Peer review of "Effects of sample size on estimation of rainfall extremes at high temperatures"

_Natural Hazards and Earth System Sciences, 2016_

## Referee Comment (RC1) · Anonymous Referee #1 · 2 Aug 2016

Journal: NHESS Title: Effects of sample size on estimation of rainfall extremes at high temperatures Author(s): B. Boessenkool et al. MS No.: nhess-2016-183 MS Type: Research article

General comments:

This paper examines the effect of sample size on the extreme precipitation-temperature relationship, often referred to as the precipitation scaling. Extreme precipitation generally increases with temperature but in some regions of the world, a reversal in the scaling is observed at higher temperatures. This reversal is generally attributed to the lack of moisture availability. Here, authors show that this reversal may also arise from a simple statistical artifact. Indeed, precipitation events at higher temperature are generally convective in nature and very localized in space (and thus often missed by the

observing network), resulting in smaller sample sizes compared to large-scale precip-
itation at lower temperatures. Authors suggest that the use of parametric quantiles
(instead of empirical quantiles) to estimate precipitation extremes at higher tempera-
tures may overcome this statistical limitation. The paper is of interest.

However, my main concern is the use of the weighted average of distributions in quan-
tile estimation. I don't see why quantile should be estimated from an average of distri-
butions that have, as authors acknowledge, inherent structural differences. Moreover,
differences in quantile estimates were found for different software packages. It seems
that the main conclusions of the paper rely heavily on the choice of distributions, the
choice of software packages and the choice of method to estimate distribution param-
eters.

Figure 8 (right panel): The spread in parametric quantile across stations at tempera-
ture >20°C is surprising. Empirical quantile on the left panel suggest that the scaling
is relatively homogeneous across stations at both low and high temperatures. The
high spread in parametric quantile at T>20°C seems suspicious and unrealistic (∼200
mm/hour!).

Technical corrections:

Figure 3: I would show only the 99 or 99.9 percentiles for clarity. Authors do not provide
any explanation for the behavior of the weighted distributions for a truncation proportion
around 0.1. Also, "other" distributions should also be removed from the figure as they
make the plot very busy and are not commented in the text.

I think authors should replace the term "quantile" by "percentile" throughout the text.
For instance, 99.9 quantile should read 99.9 percentile.

Section 2 Empirical quantile estimation. Line 29: super-CC should be defined

---

## Referee Comment (RC2) · Anonymous Referee #2 · 2 Aug 2016

Although interesting I find this paper quite confusing. It combines different distributions (GPD, Wakeby, Weibull, etc.) and different parameter estimation procedure (L-moment, maximum likelihood, etc.), different threshold values. It is difficult to come out with a clear picture of the whole think as many elements can have an impact on their conclusion that decreasing extreme rainfall below C-C scaling above 20°C can be explained by sampling issue. Many authors (e.g. Lenderink and van Meijgaard 2010; Panthou et al. 2014) used dew point temperatures to check for possible changes in relative humidity with temperature and concluded that, for some regions, decreasing intensities at higher above 20°C disappeared suggesting that humidity is a limiting factor for these regions. In these cases, sample sizes remain unchanged as the number of rainfall recorded for these temperature bins remains unchanged and therefore the change in the C-C scaling cannot be attributed to sampling issue. Therefore I would suggest

that the authors, if these records are available, use dew point temperatures to look at possible change in the shape of the extreme rainfall temperature scaling. I also have some concerns regarding some of the results. For instance in Figure 5, it is disturbing that the parametric estimates doesn't converge to the empirical value. Regarding that point, the authors mentioned on lines 26: 'Due to the inherent structural differences between distribution functions, parametric quantile estimates range from 20 to 40 mm/h, thus the weighted average is slightly higher than the empirical value of the full dataset.' This is not a convincing explanation (and not an explanation at all). This Figure is important, and for me, this discrepancy between empirical and parametric estimates cast a shadow on all the other results. Figure 6 raises also important questions. Usually L-moment estimates are less bias for small sample size than MLE. However, in this figure it seems to be the opposite. How can this be? The authors should look at the (huge) literature on the subject. The authors also refer to various R packages without any further details about the methods behind. A more complete and rigorous scientific background need to be provided on these methods. I recommend a major revisions for this paper. The authors need to a comprehensive revision of their paper. The authors need to clarify the whole methodology (please get to the point) and convince the reader that their development is free of any problems or bugs. They also need to look at C-C scaling using dew point temperature series to see if the sampling problem is still apparent there. I think that this should be minimally done to consider a possible publication of this paper.

---

## Author Comment (AC1) · 4 Oct 2016

1. The main issue raised by referee #1 is that a weighted average is computed from different distributions which might not be comparable per se.

We use a weighted average to better visualize the results from different distributions and to reduce the effect of distribution choice. We want to address the problem of choosing distributions / methods to single out unreasonable distribution choices from being included in the average. This will likely deal with the issues also pointed out by referee #1, like the suspiciously high results in figure 8 or the peak in figure 3, which stems from the Rice distribution. It might also explain why parametric quantiles in figure 5 do not converge to the 'true' value, as criticized by referee #2. As a strategy, we will inspect the sample size dependency of each distribution function and for several GPD

methods. We will repeatedly take random samples with different sample sizes from the full set of precipitation records. The 5 distributions showing the lowest dependency on sample size (the earliest convergence to the value for the full sample as per figure 6) will be selected for the weighted average. To make this selection more robust, the procedure will be repeated with different datasets.

2. Referee #2 points out that we combine many distributions, parameter estimation procedures, truncations and software packages and requests a more complete description of these methods. Referee #1 suspects that our main conclusions heavily rely on the choice of distributions, estimations, methods and R packages.

Part of our objective is to show exactly that different distributions and fitting methods yield different results, where some are more biased with sample size than others. We will try to write this in a more comprehensible way, as suggested by referee #2. Considering different methods and software packages (apart from using different distributions) has a pragmatic value. Readers should become aware that using the same distribution with the same estimation method (GPD MLE) might yield different results if different software implementations are used. This is worrying and unsurprising at the same time. We think it is beyond the scope of this paper to scrutinize all these implementations in detail, as suggested by referee #2. The main purpose is to show that sample size dependency may partly explain the drop in precipitation scaling. Our second aim is to point out that quantile estimation methods have an influence on the estimated P-T relationship. However, we will try to provide more background on the methods while remaining concise. We will try to better communicate that it matters to use distribution fitting methods and computational implementations that have the smallest possible sample size bias.

3. Referee #2 demands that we consider dew point temperatures in order to analyze whether the quantile drop is caused by moisture limitation.

In our opinion, this requirement is beyond the scope of this paper. At least for all our

analyses with synthetic data, we "know" that there is no moisture effect. The aim of this paper is to point out the potential significance of sample size artefacts. We would like to leave it to other studies to actually discriminate between sampling and moisture effects, to avoid that this article becomes overloaded.

4. Referee #2 notes that L-moment estimates are usually less biased than MLE, and claims to find the opposite relation in Figure 6.

We show that 6 out of 7 MLE implementations demonstrate exactly the expected behavior, so we cannot quite understand this concern. We will add a note on the GMLE GPD estimate, the one exception, which seems to be connected with the singular behavior for GMLE in figure 4.

5. Minor comments from referee #1 that we will almost all implement: - revise figure 3: show only one percentile, deal with the peak at a truncation value of 0.05 (comes from the rice distribution, see point 1). "Other distributions" will not be removed from figure but referenced to in the text. - replace the word "quantile" with "percentile" - define super-CC scaling - add the note that precipitation events at higher temperature are generally convective in nature and very localized in space (and thus often missed by the observing network), resulting in smaller sample sizes compared to large-scale precipitation at lower temperatures

6. A few things we would want to add to help clarify point 2: We will describe in the figure captions that gpa in figure 3 is the same method as lmomco_LM_gpa in figure 4. Figure 4 shows that GPD MLE has no increase with truncation as shown with GPD PWM/LM. We believe this to be due to the fact that a higher truncation produces smaller samples, thus GPD MLE underestimates the true value increasingly (fig 6), thus shows no increase with truncation.